# Are Bubbles the Future of Dysphagia Rehabilitation: A Systematic Review Analysing Evidence on the Use of Carbonated Liquids in Dysphagia Rehabilitation

**DOI:** 10.3390/geriatrics8010006

**Published:** 2023-01-01

**Authors:** Kathryn Price, Grace Isbister, Susannah Long, John Mirams, David Smithard

**Affiliations:** 1King’s College Hospital NHS Foundation Trust, London SE5 9RS, UK; 2Guy’s and St Thomas NHS Foundation Trust, London SE1 7EH, UK; 3Chichester PO18 8QF, UK; 4Queen Elizabeth Hospital, Lewisham and Greenwich NHS Trust, London SE9 4QH, UK; 5Centre for Exercise and Active Rehabilitation, University of Greenwich, London SE9 2HB, UK

**Keywords:** dysphagia, carbonated water, rehabilitation, quality of life

## Abstract

Background: Dysphagia poses a huge health issue in our ageing population, impacting patients psychologically and through risk of aspiration, malnutrition and airway obstruction. The use of carbonated liquids to provide sensory enhancement as a tool to stimulate neuromuscular activity in dysphagia rehabilitation remains an area with limited research. This article reviews current evidence. Method: A data search of PubMed, CINAHL, EMBASE and Cochrane was undertaken with set search terms. Abstracts were reviewed and selected by two clinicians according to inclusion criteria and papers were assessed using PRISMA methodology. Results: Selected publications (1992–2022) involved a median of 23 participants with predominantly neurogenic dysphagia. Despite the differences in study designs all used videofluroscopy (VF) to assess outcome measures except Morishita et al. who used fiberoptic endoscopic evaluation of swallow (FEES). The studies were small scale but showed encouraging results. However, there was heterogeneity between results of specific outcome measures. One study surveyed taste which was overall positively received. Conclusions: There continues to remain limited evidence to direct the use of carbonated liquids in rehabilitation of dysphagia, however its role shows some promise. The heterogeneity of not just study designs but also study participants seems to be a primary barrier. Whilst evidence is encouraging, further prospective studies standardising patient cohorts, methodologies and quantitative outcome measures must be carried out. Longitudinal studies to look at the role of carbonated liquids in secretion management is another area of potential interest. In conclusion the use of carbonated liquid in dysphagia rehabilitation may have a potential role but without firm evidence-based research, successful use in clinical practice cannot be implemented.

## 1. Introduction

Dysphagia is the medical term for swallowing difficulties. This has an array of symptoms including dry mouth, pain on swallowing, choking, coughing, the feeling of obstruction and regurgitation [1]. Dysphagia poses a huge global health issue, with one large American population-based survey suggested that the prevalence could be approximately 16% within the community [2]. Moreover, the prevalence is increased in older people within residential care with estimations of prevalence as high as 30–60% in these populations [1]. Co-morbidities associated with ageing increase the risk of dysphagia such as stroke, dementia, obstructions from head and neck cancers and other neurological conditions (e.g., multiple sclerosis and motor neuron disease). With these conditions increasing in prevalence within the UK’s ageing population, dysphagia has potential to increase the burden on individual morbidity by impacting nutrition, aspiration risk and airway obstruction. Additionally, dysphagia may also add pressure on the National Health Service through prolonged bed stays and admissions [1] and has potential to cause huge psychological sequalae. Without rehabilitation, long term dietary modifications and even progression to alternative forms of feeding such as nasogastric tube (NG) or a percutaneous endoscopic gastrostomy (PEG) can lead to a significant reduction in quality of life as well as significant psychosocial burden [3]. Individuals with swallowing deficiencies of any origin frequently require dysphagia rehabilitation. This review focuses predominantly on neurological dysphagia and dysphagia following head and neck surgery. Common procedures for head and neck tumors often result in dysphagia which requires rehabilitation with some studies showing that aspiration risk can persist up to 6 months following surgery due to abnormalities in anatomy and neurological innovation [4,5].

Techniques such as alterations to taste, temperature, texture, consistency, chemosthetic qualities or smell of food, have been proposed to provide some benefit in the management of dysphagia [6,7]. The potential of sensory enhancement techniques has been gathering evidence for decades [5] but use of carbonated liquids is not commonplace in clinical practice within the United Kingdom, and its potential for such use is unclear. Carstens et al. suggests in a review article that oral irritation by pain (noxious receptors), temperature (thermal receptors) and touch (mechanical receptors) in the oral cavity such as from CO_2_ bubbles in carbonated liquids can stimulate trigeminal neurons to stimulate swallow [8]. This is proposed to be through the mechanisms of mechanical stimulation from the bubbles and the acidity from carbonic acid [8] although precise mechanisms are still unclear. However, there seems to be a paucity of evidence in the practical use of sensory enhancement techniques with contrasting research and a lack of similarity between studies which creates further challenges. In this systematic review we have therefore reviewed recent literature to ask the question: “Does current evidence suggest a role for carbonated drinks in the management and rehabilitation of patients with dysphagia?”

## 2. Materials and Methods

PRIMSA [9] methodology was followed to structure the review (Figure 1). A search was performed by a trained professional of the following databases: PubMed, CINAHL, EMBASE and Cochrane to identify articles relevant to the research question. The following search terms were used: Carbonated water/carbonated drink, Swallowing/swallowing difficulty, swallowing disorder, dysphagia and rehabilitation. The inclusion criteria comprised of: Adult human studies only in participants with dysphagia (healthy adult studies excluded), published in peer-reviewed journals. Studies were excluded if full text was not available, articles not available in English or from grey literature (editorials, conference abstracts or clinical guidelines).

24 articles were identified from the database search. After removal of duplicates, 14 articles were included for abstract review by two researchers taking into account the inclusion and exclusion criteria. Four additional articles were added by the research team as clinically relevant to the study following additional review of relevant literature. If either researcher considered the abstract eligible, then both authors assessed the full text. In total six articles were deemed eligible for in-depth examination meeting our eligibility criteria. All studies identified were small, single-centre studies and no large randomised control trials were identified to have been completed relevant to our search criteria. The results were assimilated, and common themes were identified. These are discussed with particular attention paid to study design, outcomes and relevance to clinical practice.

## 3. Results

Six papers published between 1992 and 2022 were reviewed; all papers included are listed and described in Table 1. Three of these were from Sweden, one from the USA, one from Japan and one from Greece. Study methods varied with some using prospective and others retrospective study designs, however all were small scale single-centre studies. Overall, 159 participants were involved in these studies, with a median of 23 per study.

All studies used VF to assess participants’ swallows except Morishita et al. (2022) who used FEES [10]. However, different specific measurements were used within the swallow analysis across different studies to assess the efficiency of swallow. Most included pharyngeal transit time (PTT) amongst measurements and used rating systems to measure objective evidence of aspiration and penetration.

The results across the studies were generally positive towards the use of Carbonated thin liquids (CTL) and there was some evidence of improvement in markers of dysphagia compared to thinner fluids on swallow assessment. However, there was gross heterogeneity between the results, and although there were some positive findings, the results of different specific measurements actually conflicted between the studies. For example, Turkington et al. (2017) [7] concluded that use of CTL showed a significant reduction in penetration and aspiration compared to non-carbonated thin liquids (NCTLs) but did not see a significant reduction in PTT. Sdvarou et al. (2012) [11] also found decreased penetration and aspiration with CTL of 5 mL and 10 mL swallows, though this improvement was not present with 25 mL swallows. They also found that CTL did not improve PTT. Conversely, Larsson et al. (2017) [12] did not see an improvement in penetration and aspiration but did see a significant reduction in PTT with CTL versus NCTL. Bulow et al. (2003) [13] actually found that all of penetration, aspiration and PTT were all significantly reduced with CTL compared to NCTL. Jennings et al. (1992) [14] made no statistical breakdown of swallowing with different liquid types but commented CTLs appeared beneficial in clearing barium and excess secretions. Morishita et al. (2022) [10] was the only study which included a thickened carbonated beverage (CThL) in its study to compare to CTL, NCTL and also non-carbonated thickened liquids. It found a significantly lower PAS with CThL compared to NCTL, but no significant difference in the residue left with different liquids [10].

**Table 1 geriatrics-08-00006-t001:** Summary of articles included for review.

Author, Year	Title	Study Design	Participants	Method	Main Conclusions
Jennings et al., 1992 [14]	Swallowing problems after excision of tumors of the skull base: diagnosis and management in 12 patients.	Single centre retrospective observational study USA	N = 12 Post skull base surgery	Modified barium swallow (MBS), analysis via 2 authors. Several ½ teaspoons of different textures trialled. Use of swallowing manoeuvres	Overall 75% demonstrated evidence of aspiration. Vocal cord augmentation reduced aspiration. Carbonated beverage appeared beneficial to eliminate copious barium and secretions. No statistical breakdown of different liquids.
Bulow et al., 2003 [13]	Videoradiographic analysis of how carbonated thin liquids and thickened liquids affect the physiology of swallow in subjects with aspiration on thin liquids	Single centre prospective study Sweden	N = 40 36 were neurologically impaired and 4 were not. 19 had previously had a stroke	VFSS assessment of administration of 3 × 5 mL swallows of various textures with cued swallows	Significant reduction in penetration/aspiration and PTT with CTLs (*p* < 0.0001, *p* < 0.0001) compared to NCTL. Additionally, significantly reduced pharyngeal retention (*p* = 0.0013) with CTL compared to NCTL, and also compared to thickened liquids (*p* < 0.0001)
Sdarvou et al., 2012 [11]	Effects of carbonated liquids on oropharyngeal swallowing measures in people with neurogenic dysphagia	Single centre, Phase 1 study of treatment effect Greece	N = 17 18–80-year-olds with confirmed neurological disorder on CT/MRI, confirmed orophyngeal dysphagia with ability to tolerate VFSS and delayed pharyngeal response on NCTL seen on VFSS. Excluded peripheral nerve disorders. All participants had suffered stroke or traumatic brain injury	VFSS assessment of increasing quantities of NCTL then CTL (5-5-10-25 mL). Images analysed and subsequently 20% of studies re-analysed by two other raters to control interrater reliability. Participants rated palatability of CTL using the modified quartermaster hedonic scale.	CTL significantly decreased PEN/ASP scores of 5 and 10 mL swallows compared to NCTL (*p* = 0.028, *p* = 0.037), though not 25 mL swallows. CTL had no significant improvement in IPS, STD and PTT. CTL actually increased STD in 25 mL swallows. 58.8% participants liked or extremely liked CTL, 23.5% were not sure and 17.7% did not like CTL.
Larsson et al., 2017 [12]	Effects of carbonated liquid on swallowing dysfunction in dementia with Lewy bodies and Parkinson’s disease dementia	Single centre retrospective observational Sweden	N = 48 All participants had Lewy body dementia or Parkinson’s disease dementia and were referred from memory clinic for videofluoroscopy between 2006–2016	Retrospective analysis of previous videofluoroscopy by two authors. Patients given 3–5 mL of various textured foods. Images assessed initially with descriptive analysis, then 25 patients included in quantitative analysis	Swallow improved in 87% individuals with carbonated liquid. Significantly reduced PTT with CTL compared to thick and thin liquids (*p* = 0.014 & *p* < 0.001) The severity of retention and depth of penetration was not significantly reduced with CTL.
Turkington et al., 2019 [15]	Impact of carbonation on neurogenic dysphagia and an exploration of the clinical predictors of a response to carbonation.	Single Centre Cohort study Sweden	N = 29 Recruitment of individuals referred for VFSS with neurogenic dysphagia who could consent and had PAS score of ≥ 3 on VFSS. Varied diagnoses including stroke, progressive neurological disorders and brain injury	All participants underwent a clinical swallow examination, VFSS and genetic taste screening. With VFSS, participants prompted to take three discrete non-fixed volume sips on request, of non-carbonated liquid, then carbonated thin liquid. VFSS analysed using viduofluoroscopy dysphagia scale. VFSS interpretation by two assessors and if a disagreement then third assessor	Significant reduction (*p* = 0.01) in severity of PAS score with CTL vs. NCTL. Significant reduction in total VDS score with CTL but no reduction in PTT with CTL vs. NCTL Large variability in individual cases, some had worse PAS with CTL
Morishita et al., 2022 [10]	Effects of Carbonated Thickened Drinks on Pharyngeal Swallowing with a Flexible Endoscopic Evaluation of Swallowing in Older Patients with Oropharyngeal Dysphagia	Cross-sectional	N = 13 Adults who underwent FEES for diagnosis of dysphagia between 2021–2022 at Yokohama Izumidai Hospital.Mean age 79.6. Varied diagnoses, most common cause for dysphagia deconditioning	Patients took non-fixed volume swallows of cold (10°) CTL, NCTL, and CThL and uncarbonated thick liquids in a random order. 5 min intervals given between swallows. Swallows assessed using FEES. Secretion burden assessed using Murray secretion scale. Patient face scale was used to measure subjective difficultly in swallowing between liquids.	PAS lower with CThL than NCTL (*p* < 0.05). No significant difference in residue between liquids. No laryngeal penetration with CThL in comparison to NCTL in three patients. Subjectively swallow was reported to be easier with CThL than NCTL. Swallowing reflex initiation better with CThL compared to NCTL but no difference between CThL and thickened liquids Small sample size, varied aetiology of dysphagia

## 4. Discussion

Swallowing impairments and dysphagia cause major problems for many people on a daily basis with the potential of malnutrition, dehydration, infection and negative social sequaele. Dysphagia is a syndrome with multiple possible aetiologies [16] and therefore any management approach has to be tailored to the underlying medical (neurological/mechanical) problem. The use of CTL and CThL is one possible therapeutic option but has not been adopted into mainstream practice.

This systematic review, as with previous systematic reviews [17], has not been able to draw any firm conclusions as to the clinical benefit of CTL/CThL. The reason for this comes down to differences in methodology: prospective vs. retrospective, sample size, different aetiological process, bolus volume and strength of carbonation, and study outcome measures. All studies were single centre with small number of participants, the largest only including 48 participants.

Different studies administered different volumes of CTL to participants, i.e., Bulow et al. [13] used 5 mL sips, Sdvarou et al. [11] used incrementing volumes up to 25 mL, whereas Turkington et al. [15] and Morishita et al. [10] did not use measured volumes but participants were required to take discrete non-measured sips on demand, or as much as they could in one mouthful. The VF assessment also varied between studies with Sdravou et al. [11], Bulow et al. [12] and Turkington et al. [15] using 25 frames/s, Jennings et al. [14] and not commenting on details of VF frame speed and Larsson et al. [13] using a frame speed of only 16 frames/s which makes it possible that their results are less accurate by possibly missing some aspiration between frames. Whilst the studies focused on participants with neurogenic dysphagia and dysphagia following surgery, participants had a range of neurological diagnoses across the studies which may have affected results, and some of the participants in the studies by Bulow et al. [13] and Morishita et al. [10] did not actually have any neurological impairment, with the most common aetiology of dysphagia in the study by Morishita et al. [10] being deconditioning.

The studies showed a variability whichn changes to physiological measurements, which has also been noted in studies of health volunteers. Significant findings, that warrant further study, include the fact that in one study, swallow improved in 87% of people [13], CTL may have a benefit over thickened fluids, in one study secretions were greatly reduced [10] and in another only 17.7% did not like the taste [13].

In the studies reviewed, the make-up of the liquids did vary slightly between studies, with addition of citric acid in some studies, as well as sucralose, trisodium and citrus flavourings in the study by Morishita et al. [10], and samarin in the study by Bulow et al. [13]. It is possible that altered taste may have impacted the results. Overall, the consideration of palatability is an important point when considering management of dysphagia, especially when rehabilitation of swallow occurs to allow a patient to taste and improve quality of life. Studies investigating the use of CTL in healthy adults noted that drinks with a lower strength carbonation were preferred [18] and older adults preferred sweeter carbonated liquids [19]. Morishita et al. asked participants to rate the perceived ease of swallowing of different liquids using a “Face scale” where participants selected a face representing the ease of swallow on a scale of 1 to 5. Overall, the results found that participants found CTL and CThL easier to swallow than non-carbonated thick liquids [10].

We must also consider the variability of the study populations when considering these findings. For example, considering comparability between the results from Europe, the USA and Asia, given ethnic and cultural diversity as well as potential health inequalities and variability of services available when considering generalisability. Moreover, all the studies available in this field are small scale single centre studies with limited capacity for randomisation therefore more robust evidence is required before these findings can be considered generalisable or applied to clinical practice.

## 5. Future Work

The studies provide a glimpse as to the potential clinical benefits of CTL in the management of swallowing impairment/dysphagia. So far, research has been unable to provide a conclusive answer as to whether carbonated beverages have a place in the management of people with swallowing impairment/dysphagia because it has focused on physiological and not clinical outcomes. Future work needs to encompass

(1)Co-production with patient groups.(2)A minimum data set so that study results can be brought together to provide a larger data set.(3)Multicentre studies (single blind studies, cohort randomisation).(4)Short-, medium- and long-term outcomes (survival, dependency).(5)Simple, patient reported outcome measure, including secretion management, food intake (volume/consistency), nutritional assessments, hydration and quality of life assessment.(6)Carer stress.

CTL is being used by individuals and individual therapists, therefore standard outcome measures and the recording of anonymised data on a national or international data base should be encouraged.

## 6. Conclusions

Importantly, despite the differences between study design and lack of conformity of results, all studies reviewed suggested there may be some clinical benefit in the use of carbonated beverages in management of dysphagia. Although, the amount of evidence is small there is a suggestion that swallows are safer and secretion management improves, and consequently, until further studies are undertaken, CTL should be limited to individual patient use.

## Figures and Tables

**Figure 1 geriatrics-08-00006-f001:**
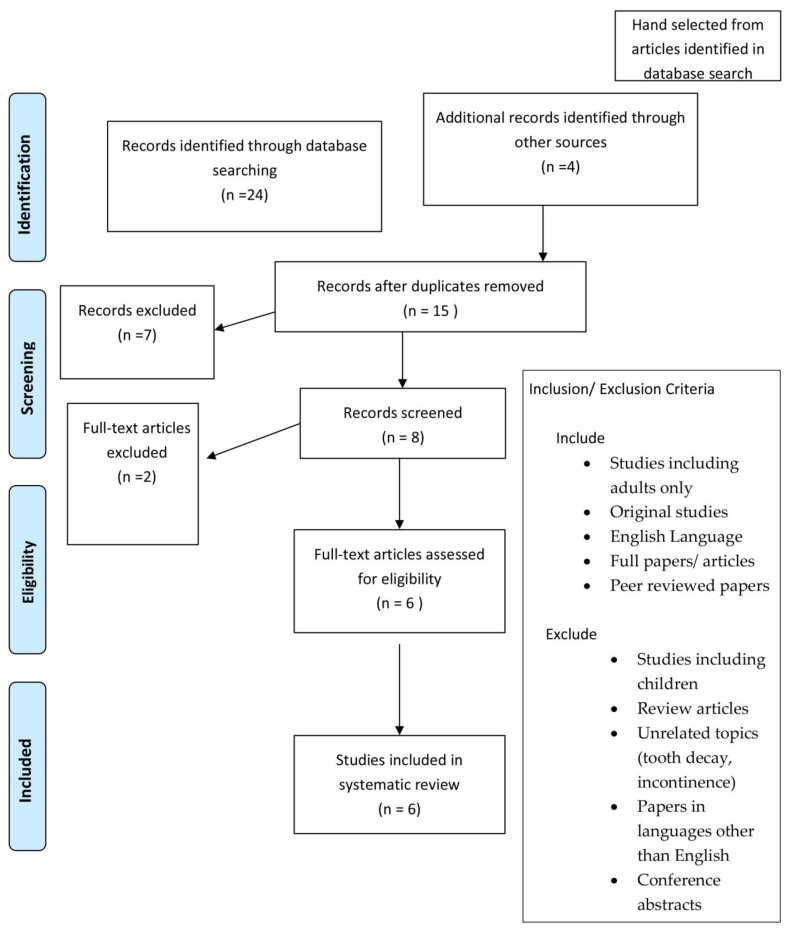
PRISMA^8^ flow diagram: systematic review papers.

## Data Availability

There is no new data in this paper. All information is available in the relevant papers cited. All search terms are documented in the paper.

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
