# Peer review of "Are Bubbles the Future of Dysphagia Rehabilitation: A Systematic Review Analysing Evidence on the Use of Carbonated Liquids in Dysphagia Rehabilitation"

_geriatrics, 2023, doi:10.3390/geriatrics8010006_

Round 1

Reviewer 1 Report

The paper submitted by Price et al. approaches a very interesting topic regarding the use of carbonated liquids in dysphagia rehabilitation.

I hope that my remarks will be useful in order to increase its’ quality.

First of all, in my eyes this paper has more the structure of a narrative review.

Line 5 – Usually „*” sign is used for corresponding authors for authors who contributed equally is recommended to use the „” sign.

Line 12 – please rephrase into „Contributed equally to the work”

Lines 37-43 – Please delete

Line 83 – Please remove „see”

Line 132 – Tables and figures should be properly introduced in the text as close as possible to the paragraph where they are cited.

Line 149 – I suggest to insert an abbreviation table at the end of the manuscript.

Line 232 – Please write the authors’ contribution in the MDPI style.

Line 236 – Please indicate if it was funded or not.

References – Please adapt all references to the MDPI style, font and size. Maybe it will be useful to try a specialized software like Zotero.

A „Perspectives” section would be useful.

I would suggest to apply all changes and try to reshape the manuscript in a much organized and reader friendly manner.

Author Response

We would like to take the opportunity to thank Reviewer ! for their comments. We read them with interest. We have made relevant changes to the manuscript and where possible have high lighted the changes in red.

The paper submitted by Price et al. approaches a very interesting topic regarding the use of carbonated liquids in dysphagia rehabilitation.

I hope that my remarks will be useful in order to increase its’ quality.

First of all, in my eyes this paper has more the structure of a narrative review.

It is difficult to respond to this comment. We folowed PRISMA guidance and structured our paper, as much as possible to follow those precepts.

Line 5 – Usually „*” sign is used for corresponding authors for authors who contributed equally is recommended to use the „†” sign.

We have ammended the signage as suggested.

Line 12 – please rephrase into „Contributed equally to the work”

This change has been made

Lines 37-43 – Please delete

This has been done

Line 83 – Please remove „see”

This has been done

Line 132 – Tables and figures should be properly introduced in the text as close as possible to the paragraph where they are cited.

We have reorganised the text to place the figures and tables as close as possible to where they are first referenced to in the text.

Line 149 – I suggest to insert an abbreviation table at the end of the manuscript.

This has ben done.

Line 232 – Please write the authors’ contribution in the MDPI style.

We have done this

Line 236 – Please indicate if it was funded or not.

This was included in the submission details. The work was not funded.

References – Please adapt all references to the MDPI style, font and size. Maybe it will be useful to try a specialized software like Zotero.

We have ammended the references.

A „Perspectives” section would be useful.

Iwe are uncertain to what is being requested here. However, the need has probably been covered in the sections on Future Work and Conclusions.

I would suggest to apply all changes and try to reshape the manuscript in a much organized and reader friendly manner.

 We have read through the manuscript, and all authors have been happy with its structure. We feel that it is readable and reader friendly. One author is a “patient” and has not commented or raised an objections to the clarity of the paper.

Reviewer 2 Report

Thank you for opportunity to review this article in Geriatrics.

This systematic review analyzing the evidence of the use of carbonated liquids in patients with dysphagia. However, there are some major points that should be attended to.

Comment to the Author

#0. How to Use This Template

Please remove this section.

#Abstruct

The authors said “but without firm evidence based research, successful use in clinical practice cannot be implemented.” Why was “Quality of life” chosen as keyword? If the carbonated water cannot be used in clinical practice, the QOL of the patients with dysphagia seems not to be changed.

#Introduction

Page2 Line 59

Percutaneous endoscopic gastronomy is a surgical method, not a measure to nutritional management. Gastrostoma or gastric fistula would be a more appropriate term.

Page2 Line 70

“but use of carbonated liquids is not commonplace in clinical practice, and its potential for such use is unclear.”

There are some study and review reporting the effect of carbonated liquid. Carbonated beverage is already used for clinical practice, therefore we consider this sentence is not appropriate.

#Results

Page3 line113

Authors abbreviated carbonated thin liquid to “CTL”.

Please add the full words of CTL in the manuscript.

Similar systematic review and meta-analysis has published in Jan 2022. Although the another study included the healthy people, the reference which should be reviewed in your study does not appear to be exhaustive.

Jennings et al reported that carbonated beverage was usable for compensatory swallowing techniques in 50% of participants. However, this study may not meet the aim of a systematic review, and there is little evidence because the evaluation whether beneficial to patients is subjective and vague.

#Figure 1 PRISMA flow diagram

The text in the blue box and the text bottom of the inclusion/exclusion criteria are cut off. Please correct the size of the text in the figure.

Author Response

We would like to take this opportunity to thank Reviewer 2 for their comments. we have responded to the questions posed and have made relevant changes to the text. Where possible we have high light date changes in red.

#0. How to Use This Template

Please remove this section.

This has been done. We apologize for the oversight at the time of submission.

#Abstruct

The authors said “but without firm evidence based research, successful use in clinical practice cannot be implemented.” Why was “Quality of life” chosen as keyword? If the carbonated water cannot be used in clinical practice, the QOL of the patients with dysphagia seems not to be changed.

Thank you for this comment. Our conclusions are that carbonated should be used in clinical practice and whilst it is used by some practitioners, it is by no means universal, particularly in the United Kingdom. From a patient perspective the main purpose of any intervention must be to improve some ones quality of life. JM is of the  personal opinion that carbonated water has markedly improved his quality of life. As such we feel it is a relevant key word.

#Introduction

Page2 Line 59

Percutaneous endoscopic gastronomy is a surgical method, not a measure to nutritional management. Gastrostoma or gastric fistula would be a more appropriate term.

Thank you for this. We would disagree. The purpose of using a nasogastric tube or a percutaneous endoscopic gastrostomy is to provide nutrition.

Page2 Line 70

“but use of carbonated liquids is not commonplace in clinical practice, and its potential for such use is unclear.”

There are some study and review reporting the effect of carbonated liquid. Carbonated beverage is already used for clinical practice, therefore we consider this sentence is not appropriate.

We have left this sentence in, but have added a qualification that, it is not common practice within the United Kingdom. We agree that some professionals across the world will include it with in their tool bag, not all do..

#Results

Page3 line113

Authors abbreviated carbonated thin liquid to “CTL”.

Please add the full words of CTL in the manuscript.

We have done this prior to subsequently abbreviating it.

Similar systematic review and meta-analysis has published in Jan 2022. Although the another study included the healthy people, the reference which should be reviewed in your study does not appear to be exhaustive.

Jennings et al reported that carbonated beverage was usable for compensatory swallowing techniques in 50% of participants. However, this study may not meet the aim of a systematic review, and there is little evidence because the evaluation whether beneficial to patients is subjective and vague.

 I am unsure as to which study is being referred to. A paper from 2022 by Morishita et al has been included in the discussion.

#Figure 1 PRISMA flow diagram

The text in the blue box and the text bottom of the inclusion/exclusion criteria are cut off. Please correct the size of the text in the figure.

This has been corrected.

Round 2

Reviewer 1 Report

Dear authors,

Thank you for re-submitting your paper.

1. You didn't fulfil all my point by point remarks.

"Line 12 – please rephrase into „Contributed equally to the work”

This change has been made"

This false. There is no modification in the text.

2. Tables are not organised in MDPI Style. Please check the template.

3.  Future work chapter should be removed and the entire text rephrased. From my point of view it's not appropriate.

Kind regards!

Author Response

We thank Reviewer ! for his time. We have answered his questions, hopefully to the satisfaction of the Reviewer and the editor.

  1. You didn't fulfil all my point by point remarks.

"Line 12 – please rephrase into „Contributed equally to the work”

We apologise for the oversight.  The three authors concerned wouldl ike to be recognised for their joint effort and hence wanted to be listed as joint first authors. Authors of papers have done this previously. The change to the text has been made as requested. The change is in red type.

  1. Tables are not organised in MDPI Style. Please check the template.

I am unable to download the template on the Mac I am using. The guide for authors includes

  1. Tables will be reformatted to the standard MDPI style prior to publication.
  2. Avoid colours. I have removed all shading.

I have not made any other changes.

  1. Future work chapter should be removed and the entire text rephrased. From my point of view it's not appropriate.

We think the last paragraph is important to suggest a future direction of research. Therefore we ask that the editor agrees to keep this section in the published paper.